# Trends of wealth-related inequality in stunting and its contributing factors among under-five children in Ethiopia: Decomposing the concentration index using Ethiopian Demographic Health Surveys 2011–2019

**Yawkal Tsega**[1]*, **Abel Endawkie**[2], **Shimels Derso Kebede**[3], **Eyob Tilahun Abeje**[2], **Ermias Bekele Enyew**[3], **Chala Daba**[4], **Lakew Asmare**[5], **Fekade Demeke Bayou**[2], **Mastewal Arefaynie**[6], **Asnakew Molla Mekonen**[1], **Abiyu Abadi Tareke**[7], **Awoke Keleb**[4], **Kaleab Mesfin Abera**[1], **Natnael Kebede**[8], **Endalkachew Mesfin Gebeyehu**[1], **Aznamariam Ayres**[2]

1 Department of Health System and Management, School of Public Health, College of Medicine and Health Sciences, Wollo University, Dessie, Ethiopia, 2 Department of Epidemiology and Biostatistics, School of Public Health, College of Medicine and Health Sciences, Wollo University, Dessie, Ethiopia, 3 Department of Health Informatics, School of Public Health, College of Medicine and Health Sciences, Wollo University, Dessie, Ethiopia, 4 Department of Environmental Health College of Medicine and Health Sciences, Wollo University, Dessie, Ethiopia, 5 Department of Epidemiology and Biostatistics, institute of Public Health, College of Medicine and Health Sciences, University of Gondar, Gondar, Ethiopia, 6 Department of Reproductive and Family Health, School of Public Health, College of Medicine and Health Sciences, Wollo University, Dessie, Ethiopia, 7 West Gondar Zonal Health Department, Amref Health Africa in Ethiopia, Gondar, Ethiopia, 8 Department of Health Promotion, School of Public Health, College of Medicine and Health Sciences, Wollo University, Dessie, Ethiopia

* yawkaltsega@gmail.com

**Data Availability Statement:** All relevant data are available within the manuscript.

## Abstract

### Background

Childhood stunting is a critical public health agenda that affects physical and cognitive development, leading to long-term health problems. Understanding its wealth related trends and contributing factors is essential for effective prospective interventions. Therefore, this study is aimed to assess the trends of childhood stunting inequality using Ethiopian Demographic Health Surveys (EDHS).

### Methods

This study employed the three consecutive EDHS datasets collected in 2011, 2016, and 2019. Socioeconomic disparity of stunting among under-five children was estimated through concentration index (CIX). Moreover, Wagstaff approach was used to decompose the relative CIX to assess the contribution of explanatory variables for the overall wealth inequality in childhood stunting.

**Funding:** The author(s) received no specific funding for this work.

**Competing interests:** The authors have declared that no competing interests exist.

**Abbreviations:** BMI, Body Mass Index; CC, Concentration index Curve; CIX, Concentration Index; DHS, Demographic and Health Survey; EDHS, Ethiopian Demographic Health Survey; EMDHS, Ethiopian Mini Demographic Health Survey; HAZ, Height-for age-z-score; WHO, World Health Organization.

## Results

The overall weighted prevalence of childhood stunting in Ethiopia was 40.76% (95%CI: 40.14%, 41.37%). The trend in the magnitude of childhood stunting decreased from 44.52% in 2011 to 37.08% in 2019. The magnitude of childhood stunting was higher (14.30%) among the poorest households than the richest households (4.70%). Moreover, the CIX of wealth inequality decreased from -0.064 in 2011 to -0.089 in 2019. Wealth index(103.38%), place of residence(34.55%), mother's education(26.73%), place of delivery(12.16%) and utilization of recommended antenatal care(12.02%) were high contributor variables in increasing the inequality, whereas administrative regions (-7.15%) and number of under-five children in the household (-4.63%) were variables contributed in the reduction of wealth inequalities in childhood stunting.

## Conclusion

This study revealed that children in the poorest households were more likely to experience childhood stunting than the children in the richest households. Factors such as wealth index, mothers education, place of residence, place of delivery, number of under-five children in the household were the contributing variables for the childhood stunting inequality. Therefore, the health decision makers better to improve the access and quality of nutritional services for the children in the poorest households in Ethiopia.

## Background

The most comprehensive measure of children's well-being and an effective way to identify inequalities in human development is the rate of linear growth. Stunting, height-for-age-z-score (HAZ) < -2SD (-2 standard deviation), among millions of children globally who not only fall short of reaching their full growth potential due to insufficient healthcare, nutrition, and disease but also experience lasting physical and cognitive harm as a result of stunted growth [1].

Malnutrition is a major universal public health problem that causes about 45% of all deaths of children, these deaths are more concentrated in low income countries countries. Stunting of under-five children is a growing health issue [2, 3]. Stunting disrupts children mental and intellectual development, and is directly associated with high morbidity and mortality rates of under-five children [4]. Documented evidence stated that an estimated 155 million children are stunted, globally. In response, the United Nation has set the Sustainable Development Goal 2.2(SDG-2.2) to end all forms of malnutrition including stunting, by 2030 [5].

Similar to other low income countries, childhood stunting is a widespread public health problem in Ethiopia and is considered a key priority area in the Health Sector Transformation Plan two (HSTP II), and the national health policy. Moreover, the government of Ethiopia with various national and international agencies working within the country have placed great emphasis on addressing the challenge of stunting among children. There has been dedicated promotion of interventions to decrease stunting such as exclusive breastfeeding, micronutrient supplementation, and many community based nutritional programs like Sekota declarations [5, 6].

Although, Ethiopia has made remarkable progress towards meeting the Sustainable Development Goals (SDGs) and decreasing undernutrition, stunting remains an important public

health problem and the prevalence is unacceptably higher so far, the latest EMDHS 2019 report stated that the prevalence of stunting was 37% [7, 8]. Various literatures have revealed that factors such as child's age, birth order, birth size, parents' education, media exposure, mother's BMI, height, nutritional knowledge and frequency of feeding, place of delivery, number of under-5 children, age of household head, household economic status and regional differentials are associated with childhood stunting [9–18].

Socioeconomic inequities in childhood stunting is increasingly attracting the attention of researchers and policymakers, thus fostering a substantial growth in the literature on health equality [16, 19]. Despite various studies conducted on stunting among under-five children, there is a noticeable gap in understanding how wealth-related inequalities influence stunting over time. Previous studies have touched on the determinants of stunting, but a comprehensive analysis focusing on the trends and contributing factors of wealth-related inequality in stunting from 2011 to 2019 is still lacking.

Therefore, this study aims to fill that gap by investigating into the trends of wealth-related inequality in stunting among under-five children in Ethiopia. By analyzing data from the Ethiopian Demographic Health Surveys conducted between 2011 and 2019, the study seeks to uncover how wealth disparities have evolved and impacted child stunting rates over these years. Furthermore, the study employed the concentration index to decompose and identify the specific factors contributing to these inequalities. This approach will provide a nuanced understanding of the underlying causes of wealth-related disparities in stunting, offering valuable insights for policymakers.

## Methods

### Data sources, study settings, and populations

This study used data from the EDHS conducted in 2011, 2016, and 2019. The Demographic Health Surveys (DHS) are national representative surveys collecting data on various health-related indicators such as fertility, infant and child mortality, reproductive age women's health, and childhood nutritional status in the country. The EDHS used a stratified two-stage clustered sampling technique in which Enumeration Areas (EAs) were selected in the first stage and 28–30 households were selected from each EAs in the second stage. The study populations for this study were under-five children (U5C) obtained from three rounds of the EDHS (2011–2019). The three EDHS datasets were appended and weighted sample of 24,595 U5C(10,040 in 2011, 9583 in 2016, and 4,972 in 2019), born 5 years before respective surveys, participated in this study [6, 7].

### Outcome variable

The outcome variable for this study was stunting among U5C. The height-for-age index was calculated according to the WHO Multicenter Growth Reference Study 2006 Child Growth Standards, where it is expressed in standard deviation (SD) units (z-score) from the median of the reference population. Height-for-age-z-score (HAZ) < -2SD was defined as stunted (Yes "labeled as 1"), and otherwise not stunted (No "labeled as 0") [18].

### Explanatory variables

The extraction of explanatory variables was conducted through reviewing prior studies that investigated their association with stunting of U5C. The identified variables were then categorized into three groups: maternal-related factors (maternal age, maternal education, maternal occupation, maternal age at first birth, place of delivery, and utilization of recommended

**Table 1. List of explanatory variables for stunting of under-five children with their respective categories (EDHS 2011–2019).**

| Variables | Categories |
|---|---|
| Mother's age | 15–24, 25–34, 35–49 |
| Age of mother at first birth | <18, 18–24, ≥25 |
| Mother's education | No education, primary, secondary and above |
| Maternal occupation | Not working, working |
| Place of delivery | Home, health facility |
| Number of ANC visits | ≤3 visits, ≥4 visits |
| Region | Urban, agrarian, pastoralist |
| Sex of household head | Male, female |
| Place of residence | Rural, urban |
| Family size | ≤4, 5–6, ≥7 |
| Wealth status of household | Poor(1), poorer, middle, richer, richest(5) |
| No of U5C | ≤2 and ≥3 |
| Age of child | <24 months, 24–59 months |
| Sex of child | Boy, girl |
| Child size at birth | Large, average, small |
| Breast milk initiation | Early initiation(<1hr), late initiation(≥1hr) |
| Duration of breastfeeding | Unfavorable(<24 months), favorable(≥24 months) |

antenatal care services), household-level variables (administrative regions, place of residence, household wealth status, sex of household head, family size, and number of under-five children), and child-related variables (child age, child sex, child size at birth, initiation of breastfeeding after birth, and duration of breastfeeding). Mother's occupation was grouped into two categories such as not working (if mothers have no any job) and working (if mothers have job either from government employer or privately). The Ethiopian administrative regions were categorized into three like Urban (comprising Addis Ababa and Dire Dawa city administrations), Agrarian (including Amhara, Oromia, SNNP, Tigray, and Harari regions), and Pastoralist (encompassing Benshangul Gumuz, Afar, Somali, and Gambella regions). Moreover, child size was categorized into large, average and small based on mother's perception during birth. These variables, along with their detail categories, are presented in Table 1 below.

## Statistical analysis

To ensure the representativeness of the sample for the entire population, all statistical analyses were weighted using the weighting variable provided by the DHS datasets. Descriptive statistics were employed to summarize frequency and percentage of the background characteristics of the study participants, and weighted prevalence, along with a 95% confidence interval, was reported. Moreover, a p-value of <0.05 was used to declare the significance of wealth related inequality of stunting.

## Wealth inequality measurement

The wealth inequality in stunting of U5C across wealth quintile categories was estimated using the concentration curve (CC) and concentration index (CIX) in their relative form (without adjustment) [20, 21]. The construction of CC involved plotting the cumulative proportion of U5C ranked by their wealth status (starting from poorest) on the x-axis, and the cumulative magnitude of stunting among U5C on the y-axis. A perfect equality is represented by a 45-degree slope depicted from northwest to northeast.

If the CC overlaps with the line of equality, it indicates that stunting is equal among U5C across their wealth category. However, if the CC moves away from the line of equality below or above, it suggests the presence of inequality in the stunting of U5C. The larger the distance between the CC and the line of equality, the greater the degree of inequality in stunting among U5C. The value of CIX is ranged from -1 to +1, where a value of 0 indicates no inequality in stunting of U5C across their socioeconomic categories.

Moreover, a positive value of CIX implies that stunting is concentrated among the higher socio-economic classes (pro-rich). Whereas, a negative CIX indicates that stunting among U5C is concentrated at lower socioeconomic groups (pro-poor). The calculation of CIX was done by using "convenient covariance" formula described by O'Donnell et al [20];

$$CIX = \frac{2}{\mu}cov(h,\ r) \tag{1}$$

Where h is the health variable, stunting among U5C in this study, μ is its mean, and r = i/N is the fractional rank of individual i in the living standards (socioeconomic variables distribution, with i = 1 for the low socioeconomic group and i = N for the highest socioeconomic group). The user-written STATA commands lorenz estimate [22] and conindex [22, 23] were used to produce CC and measure CIX, respectively.

### Decomposition of relative concentration index

The decomposition of the relative CIX was carried out to estimate important explanatory factors contributing to the total estimated socioeconomic inequality in stunting among U5C across socioeconomic categories. Wagstaff and O'Donnell [20, 23, 24] approach was used to decompose relative CIX. Wagstaff demonstrated that the CIX can be decomposed into the contributions of individual factors to total estimated socioeconomic inequality [24]. The contribution of each determinant of stunting among U5C to the overall socioeconomic inequality was determined by multiplying the determinant's sensitivity to stunting of U5C (elasticity) with the level of socioeconomic inequality associated with that determinant (CIX of the determinant). The residual component represents the portion of the CIX that cannot be explained by the determinants considered in the analysis.

## Results

### Descriptive analysis

The weighted prevalence of stunting among U5C in Ethiopia was 40.75% (95%CI: 40.14–41.37%). The descriptive analysis showed that the majority of the study participants were from rural areas (85.56%) and fall within the age range of 25–34 (~53%). Moreover, over half of mothers have no education (~67%), followed by primary education (28.89%). About 86.33% of the households were headed by male, 22.86% and 15.29% of the households were ranked as poorest and richest, respectively with regard to their wealth status. Likewise, 75.89% of mothers gave birth at home and breast milk was initiated within an hour after birth for only 63.83% of children (Table 2).

### Stunting of under-five children across Ethiopian regions

The magnitude of stunting across each Ethiopian administrative region is presented in Fig 1 below. The prevalence of stunting among under-five children ranged from 6.40% in Oromia region to 0.40% in Addis Ababa city administration.

**Table 2. Weighted descriptive analysis of explanatory variables for the stunting of under-five children in Ethiopia from 2011–2019.**

| Variables | Response | Weighted Frequency | Weighted Percent (%) |
|---|---|---|---|
| Stunting | Yes | 10,024 | 40.76 |
| | No | 14,571 | 59.24 |
| Place of residence | Urban | 3,551 | 14.44 |
| | Rural | 21,044 | 85.56 |
| Region | Urban | 411 | 1.69 |
| | Agrarian | 22,330 | 91.78 |
| | Pastoralist | 1,589 | 6.53 |
| Family size | 1–4 | 6,227 | 25.32 |
| | 5–6 | 8,835 | 35.92 |
| | $\geq$7 | 9,533 | 38.76 |
| Number of U5C | $\leq$2 | 20,420 | 83.02 |
| | $\geq$3 | \|4,175 | 16.98 |
| Mother's age | 15–24 | 5,642 | 22.94 |
| | 25–34 | 13,033 | 52.99 |
| | 35–49 | 5,920 | 24.07 |
| Mother's education | No education | 15,903 | 64.66 |
| | Primary | 7,106 | 28.89 |
| | Secondary[+] | 1,586 | 6.45 |
| Mother's occupation | Not working | 9,877 | 50.88 |
| | Working | 9,534 | 49.12 |
| Place of delivery | Home | 18,465 | 75.89 |
| | Facility | 5,867 | 24.11 |
| ANC visits | <4 | 12,157 | 70.60 |
| | $\geq$4 | 5,061 | 29.40 |
| Sex of household head | Male | 21,232 | 86.33 |
| | Female | 3,363 | 13.67 |
| Wealth status | Poorest | 5,621 | 22.86 |
| | Poorer | 5,571 | 22.65 |
| | Middle | 5,045 | 20.51 |
| | Richer | 4,597 | 18.69 |
| | Richest | 3,761 | 15.29 |
| Age of mother at first birth | <18 | 9,828 | 39.96 |
| | 18–24 | 13,147 | 53.45 |
| | $\geq$25 | 1,620 | 6.59 |
| Sex of child | Boy | 12,576 | 51.13 |
| | Girl | 12,019 | 48.87 |
| Child age in months | <24 months | 9,929 | 40.37 |
| | 24–59 months | 14,666 | 59.63 |
| Breast milk initiation | Early | 14,630 | 63.85 |
| | Late | 8,283 | 36.15 |
| Duration of breastfeeding | <24 months | 9,159 | 37.24 |
| | $\leq$24 months | 15,436 | 62.76 |
| Child size at birth | Small | 6,137 | 31.40 |
| | Average | 7,943 | 40.64 |
| | Large | 5,463 | 27.96 |

ANC: Antenatal Care, U5C: Under-five Children

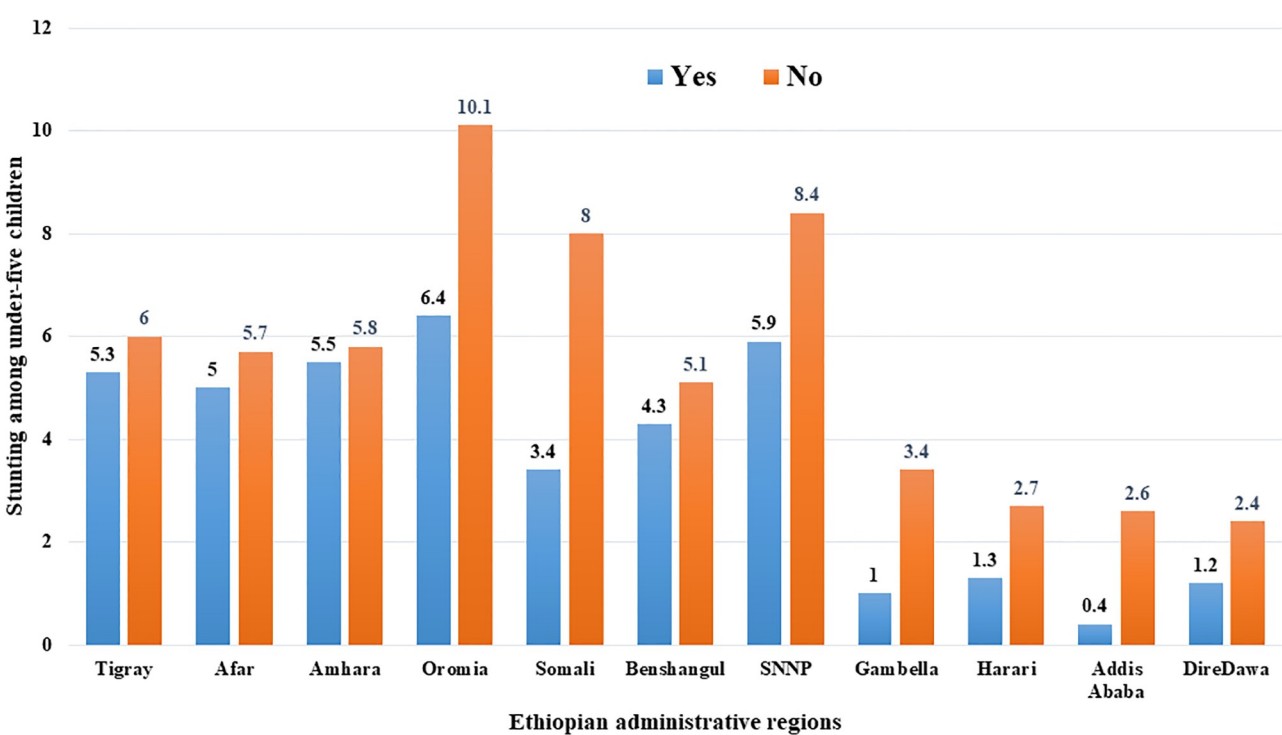

**Fig 1. The rate of stunting among under-five children across Ethiopian regions from 2011–2019.**

### Trends in the rate of stunting among under-five children in Ethiopia

The rate of stunting among U5C decreased by 7.44%, from 44.52% in 2011 to 37.08% in 2019 as presented in Fig 2. Stunting was slightly decreased by 1.64%, from 38.72% in 2016 to 37.08% in 2019.

### Trends of under-five children stunting by wealth status

Table 3 presented the prevalence of stunting among U5C across wealth categories, as determined by the household wealth index, in three consecutive survey years: 2011, 2016, and 2019. Throughout all three survey years, the rate of stunting was consistently higher among the poorest households compared to the richest households. For instance, in 2019, the rate of stunting among U5C decreased from 9.82% among the poorest households to 4.20% in the richest households. Moreover, the magnitude of stunting among U5C decreased from 11.58% in 2011 to 9.52% in 2019 among the poorest households, 1.76% difference.

### Trend changes in wealth inequality of stunting

Over the three survey years, the levels of wealth related inequalities in stunting were significantly concentrated among underprivileged (poorest) households, the CIX was negative values. The CIX was -0.064(p<0.001), -0.087(p< 0.001), and -0.089(p<0.001) in 2011, 2016 and 2019 survey years, respectively. The CIX of wealth inequalities in stunting among U5C was decreased slightly from -0.064 in 2011 to -0.089 in 2019 with the difference of -0.025 (Table 4).

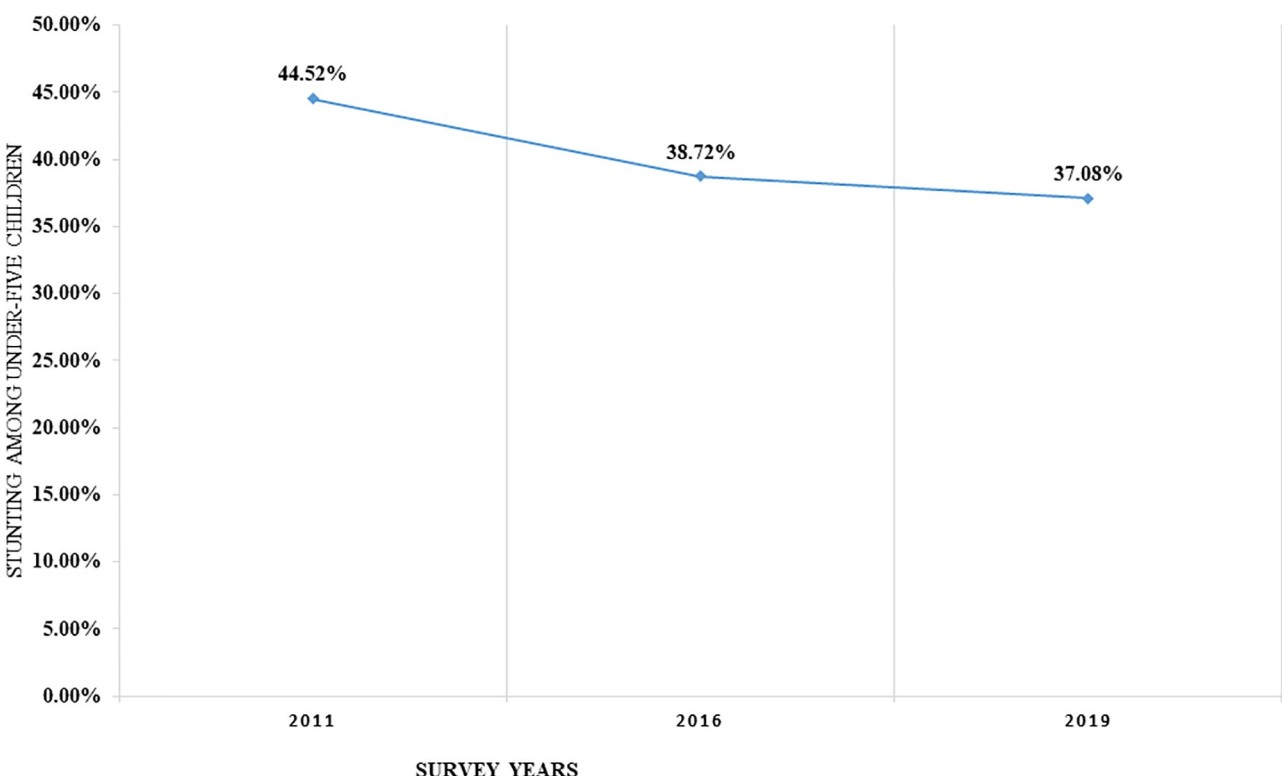

**Fig 2. Trends in the rate of stunting among under-five children in Ethiopia from 2011–2019.**

## Overall wealth inequality in childhood stunting

Fig 3 presented the rate of stunting across the five wealth categories of the households. The magnitude of stunting among U5C decreased steadily from the richest households (14.30%) to the poorest households (4.70%). This indicated that U5C in the poorest households were more likely to experience stunting than the children in the richest households.

Overall, all three survey years together, the wealth inequality concentration curve was depicted in Fig 4. The CC positioned above the line of equality, in all three survey years, revealed that higher concentration of U5C stunting among the poorest households (i.e., childhood stunting is pro-poor).

## Decomposing relative concentration index

The decomposition analysis of the relative CIX aimed to determine the contribution of important explanatory variables for the overall estimated wealth inequalities. The Table 5 below illustrates the relative contribution of each important explanatory variable to the overall estimated

**Table 3. Prevalence of stunting among under-five children across wealth status from 2011–2019.**

| Years | Household wealth status | | | | |
|---|---|---|---|---|---|
| | Poorest (n/%) | Poorer (n/%) | Middle (n/%) | Richer (n/%) | Richest (n/%) |
| 2011 | 1114(11.58) | 1063(10.58) | 970(9.66) | 893(8.89) | 430(4.28) |
| 2016 | 1000(10.43) | 973(10.16) | 764(7.97) | 620(6.47) | 353(3.69) |
| 2019 | 488(9.82) | 430(8.65) | 399(8.02) | 318(6.39) | 209(4.20) |

**Table 4. Trends of wealth related inequalities in stunting among under-five children in Ethiopia from 2011–2019.**

| Survey years | Observation | CIX | P-value | 95%CI |
|---|---|---|---|---|
| Overall | 23,540 | -0.077 | 0.000 | (-0.092, -0.062) |
| 2011 | 9,611 | -0.064 | 0.000 | (-0.083, -0.045) |
| 2016 | 8, 853 | -0.087 | 0.000 | (-0.110, -0.064) |
| 2019 | 5,076 | -0.089 | 0.000 | (-0.131, -0.047) |

CIX: Concentration index

CIX, -0.077. Each variable's contribution was depicted as a percentage and absolute value contributions. The negative percentage implies a reduction in observed inequality, while a positive percentage signifies an increase in observed inequality. The wealth status of the household was the most contributing variable in widening the inequality of stunting among U5C by 103.38% followed by place of residence and mothers' education which contributed 34.55% and 26.73%, respectively, for the observed overall inequality. However, administrative regions and the number of U5C in the household were the variables that contributed -7.15% and -4.63%, respectively, in the reduction of inequality (Table 5).

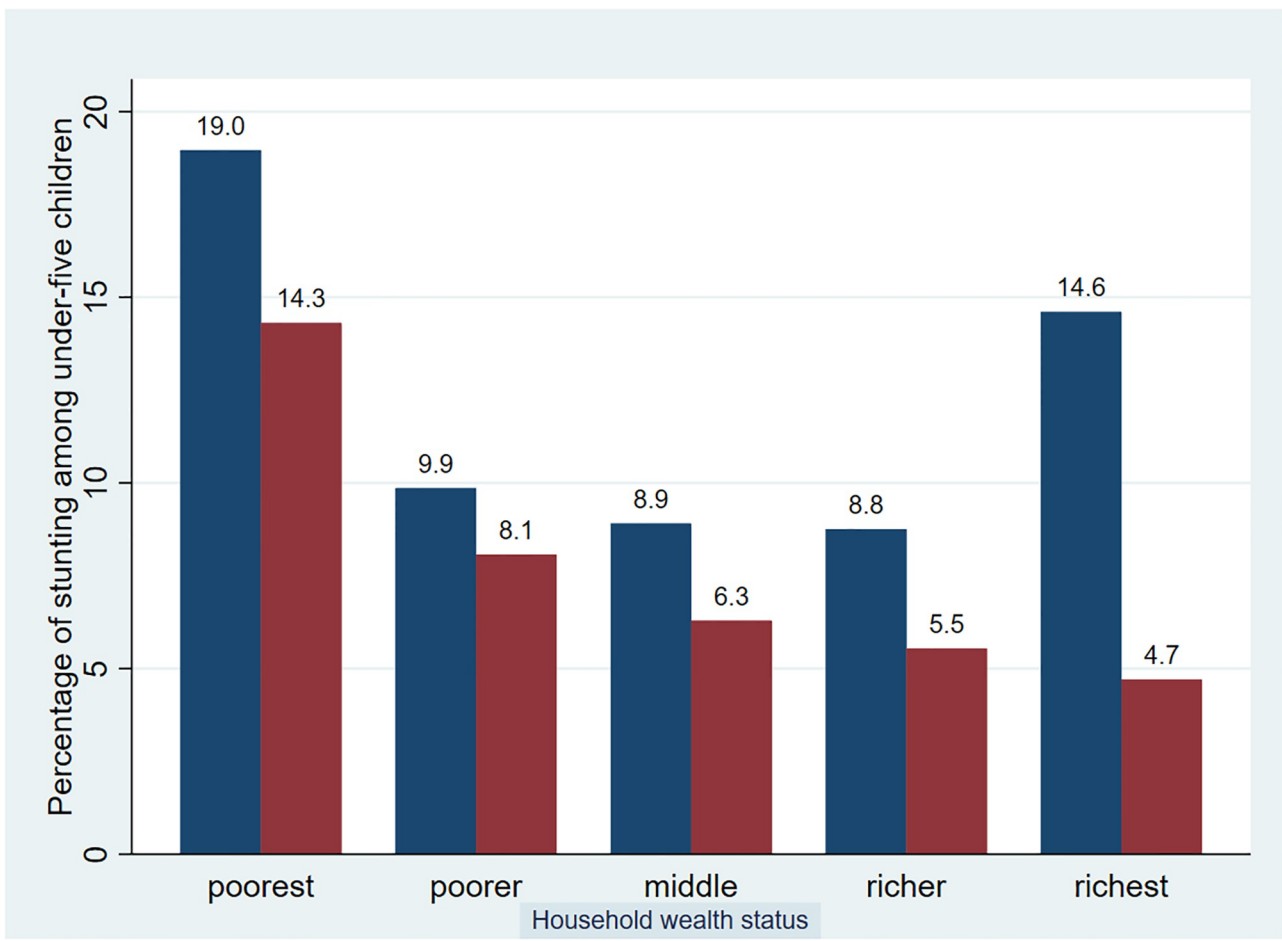

**Fig 3. Rate of stunting among under-five children across wealth categories of households in Ethiopia from 2011–2019.**

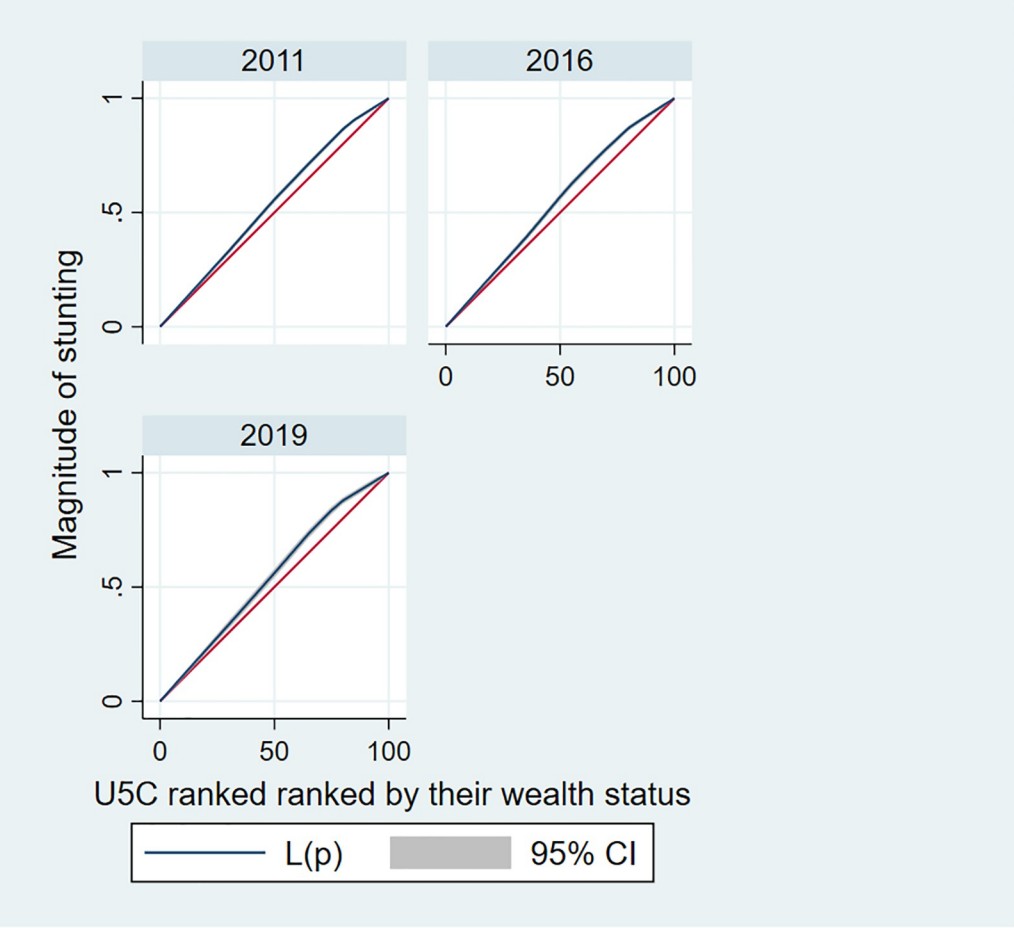

**Fig 4. Concentration curves for wealth related inequality in stunting among under-five children in Ethiopia from 2011–2019.**

## Discussion

This study aimed to assess the trend changes of wealth inequalities in stunting and its contributing factors among under-five children from 2011 to 2019 in Ethiopia. The study found that prevalence of U5C stunting was 40.76% (95%CI: 40.14%, 41.37%), decreasing from 44.52% in 2011 to 37.08% in 2019, and in all survey years stunting was concentrated among the children in the poorest households. Moreover, the study identified wealth index, mother's education, place of residence, place of delivery, and antenatal care follow up were the major contributing factors in increasing the observed wealth related inequality in childhood stunting and administrative regions and number of under-five children within the households were the variables contributed in reducing the inequality across wealth quintiles.

The finding of this study highlights the reduction in the prevalence of stunting among U5C from 2011 to 2019. The decrease from 44.56% to 37.08% suggests the Ethiopian government made substantial progress in addressing childhood stunting. However, the persistence of stunting primarily among children in poorer households indicates there is an ongoing socioeconomic disparities. The finding of the current study is consistent with previous studies conducted in Ethiopia and other low income countries [25–28]. The finding implies while the overall reduction in U5C stunting is commendable, the persistent inequality necessitates

**Table 5. Decomposition of concentration index for measuring contributions of explanatory variables for wealth inequalities in stunting of under-five children in Ethiopia from 2011–2019.**

| Variables | Elasticity | CIX | Contribution to overall [CIX = 0.14, p<0.0001] | |
|---|---|---|---|---|
| | | | Absolute contribution | Percentage contribution |
| Survey year | -0.244 | 0.001 | -0.0004 | 0.456 |
| Mother's age | -0.220 | -0.004 | 0.0010 | -1.272 |
| Mother education | -0.065 | 0.319 | -0.0206 | 26.728 |
| Mother's occupation | 0.111 | 0.010 | 0.0011 | -1.471 |
| Mother's age at 1st birth | 0.009 | 0.043 | 0.0004 | -0.499 |
| Wealth status | -0.288 | 0.277 | -0.0797 | 103.377 |
| Family size | 0.093 | -0.017 | -0.0016 | 2.017 |
| No of U5C | -0.236 | -0.015 | 0.0036 | -4.625 |
| Residence | 0.517 | -0.052 | -0.0266 | 34.547 |
| Region | -0.338 | -0.016 | 0.0055 | -7.150 |
| Sex of HH head | 0.081 | 0.003 | 0.0002 | -0.270 |
| Sex of child | -0.279 | 0.001 | -0.0002 | 0.310 |
| Child age | 0.532 | -0.008 | -0.0045 | 5.833 |
| Child size at birth | 0.454 | -0.016 | -0.0070 | 9.133 |
| Breast milk initiation | 0.008 | -0.004 | -0.0000 | 0.047 |
| Duration of breastfeeding | 0.107 | -0.002 | -0.0002 | 0.291 |
| Place of delivery | -0.027 | 0.341 | -0.0094 | 12.159 |
| ANC visit | -0.038 | 0.241 | -0.0093 | 12.015 |
| | | Explained CIX | **-0.1477** | **191.626** |
| | | Residual CIX | **0.0707** | **-91.626** |

HH: Household, CIX: Concentration index, U5C: Under-Five Children

targeted efforts to address the underlying socioeconomic factors contributing to this disparity [3, 29].

The wealth index, maternal education, and place of residence were identified as major factors contributing to wealth-related inequality in childhood stunting. This aligns with findings from other regions, such as Tanzania, where wealth and maternal education were also significant contributing factors to the stunting inequalities among under-five children. Similarly, a study in Sudan found stark inequalities in stunting across wealth quintiles, with economic status and educational inequality changing over time [30, 31].

However, factors such as Ethiopian administrative regions and the number of under-five children in a household have contributed to reducing inequality. This implied that policies and interventions at the regional level, as well as family planning and child spacing, may have positive impacts on reducing stunting disparities across socioeconomic classes. The World Health Organization also emphasizes the importance of targeting interventions to reduce stunting among disadvantaged groups, which requires a multifaceted approach including improving social services distribution, maternal education, and health infrastructure [29].

## Policy and practical implications

To address the socioeconomic disparities in childhood stunting in Ethiopia, health decision-makers should implement targeted nutritional programs for the poorest households, enhance maternal education, and improve access to health services, particularly in rural areas. Promoting and improving the utilization of antenatal and postnatal care services, along with family

planning and child spacing, can significantly reduce stunting rates. Developing localized policies and interventions tailored to regional needs, establishing robust monitoring and evaluation systems, and fostering public-private partnerships can further support these efforts. By focusing on these specific interventions, policymakers can make significant strides in reducing childhood stunting and improving overall child health outcomes in Ethiopia.

### Limitations of the study

This study utilized secondary data collected in 2011, 2016, and 2019, which may not reflect the current status of U5C stunting inequality in Ethiopia.

### Conclusion

The findings of this study revealed there is a decrement of stunting among under-five children in Ethiopia from 2011 to 2019. However, the study also uncovered that stunting remained concentrated among children from poor households throughout the three survey years. Moreover, the study identified several contributing factors for widening wealth-related inequality, including the household's wealth index, mother's education, place of residence, place of delivery, and antenatal care follow-up. Administrative regions and the number of under-five children within the household were found to have an impact on reducing the inequality across wealth quintiles. These findings suggest the need for targeted interventions that address socioeconomic disparities and prioritize the most vulnerable populations to further reduce stunting rates among Ethiopian children.

### Acknowledgments

The authors are honored to appreciate the Demographic and Health Surveys (DHS) Program for providing EDHS dataset with authorization letter.

### Author Contributions

**Conceptualization:** Yawkal Tsega, Ermias Bekele Enyew, Asnakew Molla Mekonen, Aznamariam Ayres.

**Data curation:** Yawkal Tsega, Abel Endawkie, Shimels Derso Kebede, Chala Daba, Lakew Asmare, Fekade Demeke Bayou, Mastewal Arefaynie, Asnakew Molla Mekonen, Abiyu Abadi Tareke, Awoke Keleb, Natnael Kebede, Endalkachew Mesfin Gebeyehu, Aznamariam Ayres.

**Formal analysis:** Yawkal Tsega, Aznamariam Ayres.

**Funding acquisition:** Fekade Demeke Bayou, Mastewal Arefaynie.

**Investigation:** Yawkal Tsega, Chala Daba, Lakew Asmare, Fekade Demeke Bayou, Mastewal Arefaynie, Asnakew Molla Mekonen, Abiyu Abadi Tareke, Natnael Kebede.

**Methodology:** Yawkal Tsega, Abel Endawkie, Shimels Derso Kebede, Eyob Tilahun Abeje, Ermias Bekele Enyew, Chala Daba, Lakew Asmare, Fekade Demeke Bayou, Mastewal Arefaynie, Asnakew Molla Mekonen, Abiyu Abadi Tareke, Awoke Keleb, Kaleab Mesfin Abera, Natnael Kebede, Endalkachew Mesfin Gebeyehu, Aznamariam Ayres.

**Project administration:** Yawkal Tsega, Mastewal Arefaynie, Asnakew Molla Mekonen, Abiyu Abadi Tareke, Awoke Keleb, Endalkachew Mesfin Gebeyehu.

**Resources:** Yawkal Tsega, Abel Endawkie, Shimels Derso Kebede, Eyob Tilahun Abeje, Ermias Bekele Enyew, Chala Daba, Lakew Asmare, Asnakew Molla Mekonen, Abiyu Abadi Tareke,

Awoke Keleb, Kaleab Mesfin Abera, Natnael Kebede, Endalkachew Mesfin Gebeyehu, Aznamariam Ayres.

**Software:** Yawkal Tsega, Abel Endawkie, Shimels Derso Kebede, Lakew Asmare, Aznamariam Ayres.

**Supervision:** Yawkal Tsega, Ermias Bekele Enyew, Chala Daba, Lakew Asmare, Fekade Demeke Bayou, Mastewal Arefaynie, Asnakew Molla Mekonen, Abiyu Abadi Tareke, Awoke Keleb, Kaleab Mesfin Abera, Endalkachew Mesfin Gebeyehu, Aznamariam Ayres.

**Validation:** Abel Endawkie, Eyob Tilahun Abeje, Chala Daba, Lakew Asmare, Mastewal Arefaynie, Asnakew Molla Mekonen, Awoke Keleb, Kaleab Mesfin Abera, Endalkachew Mesfin Gebeyehu.

**Visualization:** Yawkal Tsega, Abel Endawkie, Eyob Tilahun Abeje, Ermias Bekele Enyew, Chala Daba, Lakew Asmare, Awoke Keleb, Natnael Kebede.

**Writing – original draft:** Yawkal Tsega, Abel Endawkie, Shimels Derso Kebede, Ermias Bekele Enyew, Lakew Asmare.

**Writing – review & editing:** Yawkal Tsega, Abel Endawkie, Shimels Derso Kebede, Eyob Tilahun Abeje, Ermias Bekele Enyew, Chala Daba, Lakew Asmare, Fekade Demeke Bayou, Mastewal Arefaynie, Asnakew Molla Mekonen, Abiyu Abadi Tareke, Awoke Keleb, Kaleab Mesfin Abera, Natnael Kebede, Endalkachew Mesfin Gebeyehu, Aznamariam Ayres.

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
