## [Decision Letter · Decision Letter 0]

24 Jul 2024

PONE-D-24-15084Trends of wealth-related inequality in stunting among under-five children in Ethiopia: Decomposing the concentration index using Ethiopian Demographic Health Surveys 2011-2019PLOS ONE

Dear Dr. Tsega,

Thank you for submitting your manuscript to PLOS ONE. After careful consideration, we feel that it has merit but does not fully meet PLOS ONE’s publication criteria as it currently stands. Therefore, we invite you to submit a revised version of the manuscript that addresses the points raised during the review process.

Please submit your revised manuscript by Sep 07 2024 11:59PM. If you will need more time than this to complete your revisions, please reply to this message or contact the journal office at plosone@plos.org. Please include the following items when submitting your revised manuscript:A rebuttal letter that responds to each point raised by the academic editor and reviewer(s). You should upload this letter as a separate file labeled 'Response to Reviewers'.A marked-up copy of your manuscript that highlights changes made to the original version. You should upload this as a separate file labeled 'Revised Manuscript with Track Changes'.An unmarked version of your revised paper without tracked changes. You should upload this as a separate file labeled 'Manuscript'.

We look forward to receiving your revised manuscript.

Kind regards,

Jayanta Kumar Bora,PhD

Academic Editor

PLOS ONE

Journal Requirements:

Reviewers' comments:

Reviewer's Responses to Questions

**Comments to the Author**

1. Is the manuscript technically sound, and do the data support the conclusions?

Reviewer #1: Partly

Reviewer #2: Yes

2. Has the statistical analysis been performed appropriately and rigorously? 

Reviewer #1: I Don't Know

Reviewer #2: Yes

3. Have the authors made all data underlying the findings in their manuscript fully available?

Reviewer #1: No

Reviewer #2: Yes

4. Is the manuscript presented in an intelligible fashion and written in standard English?

Reviewer #1: Yes

Reviewer #2: No

5. Review Comments to the Author

Reviewer #1: • Data of 2016 not well discussed and information on 2019 on prevalence of stunting not consistent with 2019 EDHIS report of Ethiopia.

• In table 2 the variable region used to differentiate Agrarian, Urban and pastoralist, but the word “region” has different administrative meaning in Ethiopia

• How the rural resident less than Agrarian + pastoralist

• The EDHS data used in the article limited to 2011, 2016 and 2019. What was the importance of adding the other years in annexed line graph?

• Administrative region information constrained. While Ethiopia is the country with more administrative regions across geographic areas with varies stunting status, in EDHIS, this article provide constrained information about such variation which is not clearly discussed

• Writing style, font size and type needs review for consistent presentation.

• There is no discussion and possible justification for increase in stunting among urban from 2016 to 2019 EDHIS

• The Example in line # 188-189 not clearly narrated

• The conclusion not mentioned increasing trends of stunting in urban setting

Reviewer #2: Dear authors, you have researched an important topic that truly needs further exploration and intervention. Your English writing shows promise in conveying your ideas, yet could be advanced further through additional details and improved flow.

6. PLOS authors have the option to publish the peer review history of their article (what does this mean?). If published, this will include your full peer review and any attached files.

Reviewer #1: **Yes: **Tofik Abajebal

Reviewer #2: No

---

## [Author Response · Author response to Decision Letter 0]

7 Aug 2024

Dear Academic Editor and Reviewers,

We are deeply appreciative of your response to our manuscript # PONE-D-24-15084, titled "Trends of wealth-related inequality in stunting and its contributing factors among under-five children in Ethiopia: Decomposing the concentration index using Ethiopian Demographic Health Surveys 2011-2019". Your valuable comments, expert suggestions, and positive evaluation have significantly enhanced the quality of our work.

In light of the constructive feedback from the reviewers, we have meticulously revised and updated the manuscript. All comments have been addressed to improve clarity and resolve the issues raised. Furthermore, an English language expert has reviewed the manuscript to correct any grammatical inaccuracies.

These revisions have greatly enriched the manuscript. We are eager to publish the manuscript in your esteemed journal to reach a relevant audience and influence policy changes aimed at eradicating stunting in Ethiopia and similar contexts.

We would like to reiterate our profound gratitude to the academic editor and reviewers for their time and constructive feedback. We have diligently addressed all the reviewer's comments, which are attached on the following page.

Best regards,

Corresponding author: Yawkal Tsega

Department of Health System and Management, School of Public Health,

College of Medicine and Health Sciences, Wollo University, Dessie, Ethiopia.

Email: yawkaltsega@gmail.com

Mobile: +251933559351

Response to Reviewer 1 comments

Comment 1: Data of 2016 not well discussed and information on 2019 on prevalence of stunting not consistent with 2019 EDHIS report of Ethiopia.

Authors’ response: Thank you for your comment and we appreciate your feedback. We made corrections regarding with discussing about EDHS 2016 data and the prevalence of stunting in 2019. We made corrections on the points raised in the revised manuscript.

Comment 2: In table 2 the variable region used to differentiate Agrarian, Urban and pastoralist, but the word “region” has different administrative meaning in Ethiopia. Comment 3: How the rural resident less than Agrarian + pastoralist?

Authors’ response for Comment 2-3: We appreciate your insightful comments. Indeed, you’re correct that ‘region’ refers to the administrative divisions as per the Ethiopian government. However, we employed the variable ‘administrative region’ and categorized it into Urban (comprising Addis Ababa and DireDawa city administrations), Agrarian (including Amhara, Oromia, SNNP, Tigray, and Harari regions), and Pastoralist (encompassing Benshangul Gumuz, Afar, Somali, and Gambella regions). To avoid any confusion for readers, we have operationalized these variables in the revised manuscript.

Comment 4: The EDHS data used in the article limited to 2011, 2016 and 2019. What was the importance of adding the other years in annexed line graph?

Authors’ response: Thank you for your comment and feedback. We have chosen to use the EDHS datasets from 2011, 2016, and 2019 to focus on the most recent decade. This choice allows us to analyze the most current trends in wealth-related inequality of stunting among under-five children in Ethiopia. The authors' primary analysis is focused on the recent decade, as this is the period most relevant to current policy and intervention planning. 

Comment 5: Administrative region information constrained. While Ethiopia is the country with more administrative regions across geographic areas with varies stunting status, in EDHIS, this article provide constrained information about such variation which is not clearly discussed

Authors’ response: Thank you for your insightful comments. We depicted the variation in the rate of stunting among under-five children across the Ethiopian administrative regions through bar graph as presented in Fig 1 in the revised manuscript. 

Comment 6: Writing style, font size and type needs review for consistent presentation.

Authors’ response: The authors would like to express their gratitude for your expert advice. Your suggestions have been instrumental in our comprehensive revision of the manuscript's writing style, font size, and typographical errors. We have taken your recommendations into account in the revised manuscript.

Comment 7: There is no discussion and possible justification for increase in stunting among urban from 2016 to 2019 EDHIS

Authors’ response: Thank you for your comment. You’re right that discussing and justifying the increase of stunting in urban areas is crucial for health decision makers. However, the aim of our study was assessing the trends of wealth related inequality in stunting among under-five children in Ethiopia. This study didn’t assess the trends of stunting in urban area specifically. 

Comment 8: The Example in line # 188-189 not clearly narrated

Authors’ response: Thank you so much for your comments and bringing this point to our attention. We addressed your suggestions and made clarity in the revised manuscript. 

Comment 9: The conclusion not mentioned increasing trends of stunting in urban setting

Authors’ response: We appreciate insightful comments. The aim of this study was assessing the trends of wealth related inequality in stunting among under-five children in Ethiopia. This study didn’t assess the trends of stunting in urban area specifically. 

Response to reviewer 2 comments. 

General comment: Dear authors, you have researched an important topic that truly needs further exploration and intervention. Your English writing shows promise in conveying your ideas, yet could be advanced further through additional details and improved flow.

Authors’ response: Dear Reviewer 2, thank you for your constructive feedback and kind words about our manuscript. We appreciate your suggestion to enhance the manuscript by providing additional details and improving the flow of our writing. We certainly took these points into consideration in our revision.

Under title 

Comment 1: The authors could consider crafting a more comprehensive title that encompasses both the prevalence and explanatory variables.

Authors’ response: We appreciate your insightful comments. We made corrections of the research topic in the revised manuscript through taking your suggestion into consideration. 

Under abstract 

Comment 1: In the background section, it is important for the authors to clearly establish the connection between the study's objective and the research gap, specifically addressing the need for decomposing concentration. Additionally, including confidence intervals and margins of error in the method section would be valuable.

Authors’ response: Thank you for your insightful comments. We agree that establishing a clear connection between the study's objective and the identified research gap is crucial. In the revised manuscript, we addressed your comments. 

Under Introduction 

Comment 1: To improve the flow of the introduction, the authors could consider starting with a discussion of the situation of stunting and then transitioning into how it can be measured and evaluated. 

Authors’ response: Thank you for your comment. We revised the introduction part of the manuscript taking your comments into consideration.

Under Materials and methods

Comment 1: Certain explanatory variables require clarification, such as providing definitions or additional information. For example, it would be helpful to specify the meaning of "not/working" under maternal occupation and provide clear criteria for categorizing Child size at birth as large, medium, or small. 

Authors’ response: Thank you for your comments and suggestions. We operationalized the some of the explanatory variables as per your suggestion. 

Comment 2: Additionally, indicating the specific time frames for late and early initiation of breastfeeding and providing the values of the wealth index for the poorest and richest categories would enhance clarity. Furthermore, it is important to explicitly state the objectives of decomposing the relative concentration index in the methodological section.

Authors’ response: Thank you for your valuable suggestions. Based on your comments and suggestions, the manuscript was revised through addressing your comments and suggestions. 

Under Report/result

Comment 1: Some reports looks discussion 

Authors’ response: Thank you for your comment. We revised the overall result part of the manuscript as per your suggestions. 

Comment 2: It is essential to exercise caution when using the wealth index and socio-demographic factors 

Authors’ response: Dear Reviewer 2, we appreciate your suggestions. We took your suggestions into consideration in the revised manuscript. 

Under Discussion 

Comment 1: In line 242, the conclusion is made without providing reasoning for the reduction of stunting, which may leave readers expecting further explanation. 

Authors’ response: Thank you so much for your comment. We revised the manuscript and addressed your comments in the revised manuscript. 

Comment 2: To facilitate a clearer understanding for drawing conclusions and making recommendations, it would be beneficial to discuss the role of each explanatory variable independently, rather than summarizing them together in line 249.

Authors’ response: We sincerely value your feedback and recommendations. With your suggestions in mind, we have restructured the discussion section to include the policy implications and practical applications of our findings.

Under Conclusion:

Comment 1: Given the authors' emphasis on prevalence in the report and conclusion, it would be advisable to either focus on prevalence in the title, objective, and method sections as well, or temper its significance in the results and conclusion.

Authors’ response: Thank you for your comment. We revised the conclusion section of the manuscript as per your comments. 

Under References:

Comment 1: No comments.

Authors’ response: Thank you so much for your revision on our manuscript.

---

## [Decision Letter · Decision Letter 1]

15 Oct 2024

PONE-D-24-15084R1Trends of wealth-related inequality in stunting and its contributing factors among under-five children in Ethiopia: Decomposing the concentration index using Ethiopian Demographic Health Surveys 2011-2019PLOS ONE

Dear Dr. Tsega,

Thank you for submitting your manuscript to PLOS ONE. After careful consideration, we feel that it has merit but does not fully meet PLOS ONE’s publication criteria as it currently stands. Therefore, we invite you to submit a revised version of the manuscript that addresses the points raised during the review process.

We look forward to receiving your revised manuscript.

Kind regards,

Satyajit Kundu

Academic Editor

PLOS ONE

**Additional Editor Comments:**

After reviewing the revised manuscript, I strongly have some suggestion before reaching to a decision. The authors sometimes argued in the rebuttal letter that they have addressed the comment in the revised manuscript, but they didn't. My specific suggestion are given below:

-Comment from reviewer 1: "There is no discussion and possible justification for increase in stunting among urban from 2016 to 2019 EDHIS" - Thought this is not your primary objective, it is a significant finding. I would suggest to discuss about this change and also mention in the conclusion.

-Comment from Reviewer 2: "In the background section, it is important for the authors to clearly establish the connection between the study's objective and the research gap, specifically addressing the need

for decomposing concentration" - The author respond that they have addressed it, but I didn't notice. A clear understanding of whats already know on this topic, what are the gaps, and how this study adds value, is very important in the background section.

-What was you response against this comment? "Additionally, including confidence intervals and margins of error in the method section would be valuable"

-If you have valid points, you can deny to address all the comments from reviewers. But for that you have to clearly depict your points of argument for both accepting and denying to address the comments. I strongly suggest you to read all the comment specifically and try to address them and make a rebuttal letter with your argument why you didn't address any comments.

Reviewers' comments:

Reviewer's Responses to Questions

**Comments to the Author**

1. If the authors have adequately addressed your comments raised in a previous round of review and you feel that this manuscript is now acceptable for publication, you may indicate that here to bypass the “Comments to the Author” section, enter your conflict of interest statement in the “Confidential to Editor” section, and submit your "Accept" recommendation.

Reviewer #2: All comments have been addressed

2. Is the manuscript technically sound, and do the data support the conclusions?

Reviewer #2: Yes

3. Has the statistical analysis been performed appropriately and rigorously? 

Reviewer #2: Yes

4. Have the authors made all data underlying the findings in their manuscript fully available?

Reviewer #2: Yes

5. Is the manuscript presented in an intelligible fashion and written in standard English?

Reviewer #2: Yes

6. Review Comments to the Author

Reviewer #2: Dear authors, I am pleased with the quality of your work, and I believe it makes a valuable contribution to the field.

7. PLOS authors have the option to publish the peer review history of their article (what does this mean?). If published, this will include your full peer review and any attached files.

Reviewer #2: No

---

## [Author Response · Author response to Decision Letter 1]

4 Nov 2024

Date: November 04, 2024

To: PLOS ONE 

Subject: Submission of Revised Manuscript

Dear Editor and Reviewers,

We appreciate your feedbacks to our manuscript # PONE-D-24-15084R1, titled "Trends of wealth-related inequality in stunting and its contributing factors among under-five children in Ethiopia: Decomposing the concentration index using Ethiopian Demographic Health Surveys 2011-2019". Your valuable comments, and expert suggestions have significantly enhanced the quality of our manuscript.

In light of the constructive feedbacks, we have meticulously revised and updated the manuscript for second time. Furthermore, an English language expert has reviewed the manuscript to correct any grammatical inaccuracies.

We are eager to publish the manuscript in your reputable journal, PLOS ONE, to reach a relevant audience and influence policy changes aimed at eradicating stunting in Ethiopia and other similar settings

Once again, we would like to reiterate our profound gratitude to the editor and reviewers for your time and constructive feedback. We have provided responses to all comments point by point below. 

Best regards,

Yawkal Tsega

Corresponding author

On behalf of the authors

Email: yawkaltsega@gmail.com

Mobile: +251933559351

Response to Editor’s comments

Comment 1: After reviewing the revised manuscript, I strongly have some suggestion before reaching to a decision. The authors sometimes argued in the rebuttal letter that they have addressed the comment in the revised manuscript, but they didn't. My specific suggestion are given below:

Authors’ response: Dear Editor, thank you so much for your comments and we appreciate your feedback. We tried address all the comments in the revised Manuscript.

Comment 2: Comment from reviewer 1: "There is no discussion and possible justification for increase in stunting among urban from 2016 to 2019 EDHIS" - Thought this is not your primary objective, it is a significant finding. I would suggest to discuss about this change and also mention in the conclusion.

Authors’ response: Thank you for bringing this point to our attention. However, in this study, we did not analyse the prevalence of stunting in urban areas separately. There are no findings from this study indicating an increase in stunting in urban areas from 2016 to 2019. In fact, the prevalence of stunting among under-five children slightly decreased from 25.4% in 2016 to 22% in 2019.

Comment 3: Comment from Reviewer 2: "In the background section, it is important for the authors to clearly establish the connection between the study's objective and the research gap, specifically addressing the need for decomposing concentration." The author respond that they have addressed it, but I didn't notice. A clear understanding of what’s already know on this topic, what are the gaps, and how this study adds value, is very important in the background section. 

Authors’ response: Thank you for your insightful comments. We have taken the reviewers' and your feedback into consideration and have revised the background section of the manuscript. We have improved the connection between the study's objective and the research gap, clearly establishing what is already known on this topic, identifying the gaps, and demonstrating how this study adds value. Indicated from line number 85 to 97. 

Comment 4: What was your response against this comment? "Additionally, including confidence intervals and margins of error in the method section would be valuable"

Authors’ response: We appreciate your insightful comments. Your feedback has been instrumental in improving the quality of the manuscript. In this study, we used secondary data from the EDHS, so the margin of error is not needed. The EDHS employed a two-stage sampling method: in the first stage, enumeration areas were selected, and in the second stage, 28-30 households were chosen. Therefore, the margin of error is not required to calculate the sample size. However, we have included the 95% confidence interval in the methods statistical analysis subsection of this study.

Comment 5: If you have valid points, you can deny to address all the comments from reviewers. But for that you have to clearly depict your points of argument for both accepting and denying to address the comments. I strongly suggest you to read all the comment specifically and try to address them and make a rebuttal letter with your argument why you didn't address any comments. 

Authors’ response: Authors’ response: Dear Editor, we have addressed all the comments and clarified some points. We have proofread the entire manuscript and made efforts to improve its quality.

Response to Reviewer 2 feedback

Comment 1: All comments have been addressed

Authors’ response: Thank you for your positive evaluation of our manuscript and we appreciate your feedback. 

Comment 2: Dear authors, I am pleased with the quality of your work, and I believe it makes a valuable contribution to the field.

Authors’ response: Thank you so much for your kind words.

---

## [Editor Report · Decision Letter 2]

14 Nov 2024

Trends of wealth-related inequality in stunting and its contributing factors among under-five children in Ethiopia: Decomposing the concentration index using Ethiopian Demographic Health Surveys 2011-2019

PONE-D-24-15084R2

Dear Dr. Yawkal Tsega,

We’re pleased to inform you that your manuscript has been judged scientifically suitable for publication and will be formally accepted for publication once it meets all outstanding technical requirements.

Kind regards,

Satyajit Kundu

Academic Editor

PLOS ONE

---

## [Editor Report · Acceptance letter]

18 Nov 2024

PONE-D-24-15084R2 

PLOS ONE

Dear Dr. Tsega, 

I'm pleased to inform you that your manuscript has been deemed suitable for publication in PLOS ONE. Congratulations! Your manuscript is now being handed over to our production team.

Kind regards, 

on behalf of

Satyajit Kundu 

Academic Editor

PLOS ONE